# Bridging the Digital Divide: Performance Variation across Socio-Economic Factors in Vision-Language Models

**Joan Nwatu**[$]    **Oana Ignat**[$]    **Rada Mihalcea**
University of Michigan - Ann Arbor, USA
*{jnwatu, oignat, mihalcea} @umich.edu*

## Abstract

Despite the impressive performance of current AI models reported across various tasks, performance reports often do not include evaluations of how these models perform on the specific groups that will be impacted by these technologies. Among the minority groups underrepresented in AI, data from low-income households are often overlooked in data collection and model evaluation. We evaluate the performance of a state-of-the-art vision-language model (CLIP) on a geo-diverse dataset containing household images associated with different income values (Dollar Street) and show that performance inequality exists among households of different income levels. Our results indicate that performance for the poorer groups is consistently lower than the wealthier groups across various topics and countries. We highlight insights that can help mitigate these issues and propose actionable steps for economic-level inclusive AI development. Code is available at Analysis for Bridging the Digital Divide.

## 1 Introduction

The impact of AI on the general public is rapidly growing, now getting within reach of people *worldwide*. More than ever, it is critical that these models work well for *everyone*. Language and vision models are also expanding, with some already being used as foundation models (Bommasani et al., 2022), as these models are trained on enormous datasets and have been shown to possess impressive capabilities on downstream tasks across various domains.

However, since most research that yielded foundation models comes from top companies in the Western tech industry, it is unsurprising that these models tend to be biased and perform unequally

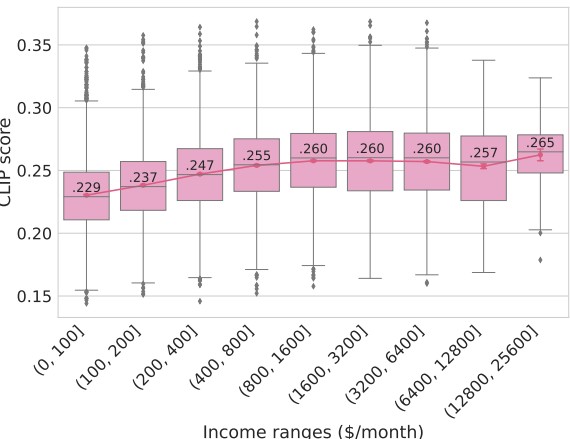

Figure 1: Median CLIP (Radford et al., 2021) alignment scores across images in Dollar Street (Rojas et al., 2022) from different *income ranges*, together with average CLIP scores with confidence values for each range. We measure a trend of increasing CLIP scores as the income range increases.

for the global population – a consequence of training AI models with data that reflects a one-sided view of the world (Buolamwini and Gebru, 2018). The rising concern about the disparate impact of AI technologies on different members of the general public has led to research investigating cases where AI models do not work well for different underrepresented groups (Bolukbasi et al., 2016; Gebru, 2020; Cirillo et al., 2020; Ahn et al., 2022; Shrestha and Das, 2022). While there is a sizable literature studying the disparate AI impact on people of different races and genders, less emphasis is placed on investigating the interaction between AI model performance and economic inequality in the world, even as research inferring economic status from images suggests that the differences between the rich and the poor can be captured by AI models (Acharya et al., 2017; Gebru et al., 2017; Yeh et al., 2020a; Machicao et al., 2022).

The consequences are prominent, as neglecting the AI impact on people of different socio-

---

[$]Joan Nwatu and Oana Ignat contributed equally to the manuscript. Rada Mihalcea initiated and guided the work, and provided overall supervision.

economic levels, is further widening the "digital divide", by excluding low-income background people from benefiting from AI applications (Lutz, 2019; Carter et al., 2020; Kitsara, 2022; Khowaja et al., 2023). As technological progress threatens to widen the economic gap between the rich and the poor, it is essential to have a clear understanding of how state-of-the-art models perform across all income levels, to help these models achieve good performance across all economic levels (Miailhe and Hodes, 2017; Khowaja et al., 2023). To address the lack of research on evaluation across economic levels, we conduct an in-depth performance evaluation of a state-of-the-art vision-language model on images from diverse household incomes. Based on our findings, we propose a series of actionable steps to "democratize" vision-language models and ensure everyone benefits from the upcoming AI revolution.

We summarize our contributions as follows. First, **we demonstrate the disparate performance of a vision-language foundation model** across groups of different economic levels. By formulating a series of research questions, we perform fine-grained analyses to identify topics and countries that require more research attention and highlight the possible issues causing difficulty for vision-language models. Second, **we uncover visual similarity analogies across countries and incomes** from diverse household appearances. Lastly, based on insights from our analyses, **we present six actionable recommendations to improve equality in vision-language model performance** across different socio-economic groups.

## 2 Related Work

**Measuring the disparate impact of AI across cultures.** Existing research on the evaluation of vision and language models has led to the discovery of unequal performance across various factors like gender, language, and race. Further analyses of these models reveal that they are mainly trained on data collected from the Western world. For instance, the concentration of NLP research on English and a handful of other languages has contributed to the inequality in developing language technologies for multiple NLP tasks across the world's languages (Blasi et al., 2022). Current NLP models yield lower performance on language tasks for non-Western languages (Hu et al., 2020; Khanuja et al., 2023).

In computer vision (CV) research, classifiers trained on popular datasets such as ImageNet (Deng et al., 2009) or OpenImages (Krasin et al., 2017) yield frequent misclassifications due to geolocation (Shankar et al., 2017). Analyses of these datasets show that 60% of the data comes from only 6 (out of 195) [$] countries, all from North America and Europe (Shankar et al., 2017). Similarly, research by De Vries et al. (2019) investigates how CV models perform on diverse, cross-cultural images from the Dollar Street dataset (Rojas et al., 2022). This work is most similar to ours; however, their focus is on object-recognition models, while ours is on a vision-language foundation model, CLIP (Radford et al., 2021). Their main finding, four years ago, is consistent with ours: *"The systems perform relatively poorly on household items that commonly occur in countries with a low household income,"* revealing that current CV models still work significantly worse for people with low-income households.

**Improving Cultural Representation in AI.** While transfer learning techniques (Ruder et al., 2019; Rahimi et al., 2019; Conneau et al., 2020) shine a ray of hope for increasing language diversity, Joshi et al. (2020) challenge the optimism towards transfer learning for multilingual NLP by highlighting that many low-resource languages contain typological features not adequately represented in richer resource languages like English. Other efforts revolve around data collection (Alahmadi et al., 2020; Koto et al., 2020; Augustyniak et al., 2022; Marreddy et al., 2022), which improve data collection techniques for low-resource languages and facilitate the participation of indigenous people in data collection for NLP research.

Using pre-trained models for vision tasks is common practice (Donahue et al., 2014; Girshick et al., 2014; Wang and Russakovsky, 2023). However, Salman et al. (2022) indicate that bias from pre-trained models can be transferred over to downstream tasks even if datasets for fine-tuning are explicitly debiased. At the same time, Wang and Russakovsky (2023) show that biases due to spurious correlations and under-representation can be counteracted through targeted dataset manipulations. In cases of under-representation, they recommend adjusting the proportion of positive labels belonging to the under-represented group and note that this process involves further collecting under-

---

[$] https://www.worldometers.info/

represented data and carefully curating datasets for fine-tuning. Similarly, Ramaswamy et al. (2023) show that adding geo-diverse data to the training dataset increases model performance.

Work toward increasing representation in CV datasets include GeoDE (Ramaswamy et al., 2023) - one of the most geo-diverse datasets, GeoYFCC (Dubey et al., 2021) - less diverse, with data mainly from Europe, Segment Anything (Kirillov et al., 2023) - a large geo-diverse segmentation dataset, and Dollar Street dataset (Rojas et al., 2022) which we use in this project because it contains income information and everyday human actions and objects.

**Using Images to Infer Economic Information.** Research work on making use of image data to identify the income of households or neighborhoods in the US includes Acharya et al. (2017) and Gebru et al. (2017), who identify low-income neighborhoods using Google Street View images.[$] They demonstrate the use of machine learning to predict the economic welfare of households as a potential alternative to traditional methods like wealth surveys. Xie et al. (2016) and Yeh et al. (2020b) also use deep learning to detect predictive wealth features in the day and nighttime satellite imagery.

Following previous work in the social sciences (Rosling et al., 2019) that demonstrated how income cuts across cultures/countries, the creators of the Dollar Street dataset point to how household appearances seem to differ especially across income groups and not necessarily across countries, as commonly believed.[$]

## 3 Methodology

We present the dataset, Dollar Street (Rojas et al., 2022) and the vision-language state-of-the-art model, CLIP (Radford et al., 2021), which we use for our experiments. By formulating and answering a series of research questions, we perform a fine-grained performance analysis of CLIP on a socio-economically diverse dataset across topics, income levels, and countries. The resulting insights lead to actionable steps to improve vision-language models' performance across diverse incomes.

### 3.1 Dollar Street Dataset

The Dollar Street dataset contains $38,479$ images collected from homes in 63 countries on four con-

tinents. The images capture everyday household items (e.g., "toothbrush", "toilet paper", "clothes"), which are called **topics**. While image resolution and size vary slightly across locations, relevant metrics such as mean and median are similar; therefore, a CV model will likely not be impacted by image resolution and size. All the images in the Dollar Street dataset have respective household socio-economic information, i.e., **income** and **location**.

**Topic Representation.** Each image is manually annotated with one or more related, textual topics: e.g., "adding spices to food while cooking", "spices". There are 291 unique topics, out of which we remove nineteen subjective topics following the work of De Vries et al. (2019) (e.g., "most loved item", "things I wish I had"). All the subjective topics are found in Appendix A.

**Income Representation.** The Dollar Street dataset contains images from homes with monthly incomes ranging from $26.9\$$ to $19,671.0\$$. The household income is calculated as consumption over an extended period (a year), expressed per adult equivalent, using the OECD (Organisation for Economic Co-operation and Development) modified scale [$], then further divided and displayed to reflect monthly consumption. This number is derived from the household's self-reported consumption and income levels. The total consumption is measured in U.S. dollars, adjusted for purchasing power parity, to account for the varied cost of living among different countries. Further information regarding the calculations can be found on the Gapminder website.[$]

We further group the income values into *geometric ranges* and *quartiles*, as described in Rojas et al. (2022). The quartile binning method, in Table 1, divides the distribution of images into an approximately equal number of images per bin, allowing for fair comparisons between the bins.

**Location Representation.** The dataset contains images from four continents: Africa, America, Asia, and Europe, and 63 countries out of the 195 that exist worldwide. The number of images for a given country ranges from $45$ in Canada to $4,704$ in India, with a median of $407$ images per country.

---

[$]https://www.google.com/maps/
[$]https://www.gapminder.org

[$]http://www.oecd.org/eco/growth/OECD-Note-EquivalenceScales.pdf
[$]https://www.gapminder.org/dollar-street

## 3.2 State-of-the-art Vision-Language Model

For our evaluation, we choose CLIP, as opposed to other language-vision models, due to its vast popularity as a foundation model (Bommasani et al., 2022), i.e., its use in a multitude of models and its impressive zero-shot performance across various tasks and datasets, e.g., text-to-image retrieval, image question answering, human action segmentation, image-sentence alignment – (Cafagna et al., 2021). However, we observe these datasets contain mostly images from North America and Western Europe, and, to the best of our knowledge, we are the first to evaluate CLIP on more diverse data.

We use the CLIP model to represent all the topics and their corresponding images. We use the pretrained Vision Transformer model ViT-B/32 (Dosovitskiy et al., 2021) to encode the images and the textual information. When computing CLIP textual topic embeddings, we concatenate the topics with given prompts (e.g., "This is a photo of *a toilet*"), as described in Radford et al. (2021). We then compute the CLIP alignment scores as the cosine similarity between the representations of the topics and their respective images. We choose to use absolute CLIP scores in our experiments because they have been shown to provide a strong signal on the relevance that an image has to a given topic (e.g., the creators of the widely used LAION dataset (Schuhmann et al., 2021) performed human evaluations to choose a CLIP threshold of 0.30 to determine label-image relevance).

## 4 Research Questions

**RQ1. Does CLIP show varying performance based on different income levels associated with the images?** Analyses of CLIP scores aggregated across income level groupings provide convincing evidence that CLIP's performance varies across different income levels.

We measure the association between CLIP scores and household images in our dataset. Using Spearman's Rank Correlation coefficient, we find a correlation of $0.35$, which indicates a *moderate association* between CLIP performance and income.

Figure 1 shows a trend of increasing CLIP scores as income range increases: performance is significantly worse on images from poorer households, while it peaks on images from upper-middle income class. Specifically, CLIP performs poorly (score $< 0.25$) on about $75\%$ of images from the

lowest income bin – bin which currently represents around $20\%$ of the world's population.[$]

| Quartile name | Income range | CLIP scores | | |
|---|---|---|---|---|
| | | ViT-B 32 | ViT-L 14 | ViT-G 14 |
| poor | 26.9 - 195.0 | 0.233 | 0.259 | 0.321 |
| low-mid | 195.4 - 685.0 | 0.250 | 0.282 | 0.350 |
| up-mid | 694.0 - 1,998.0 | 0.257 | 0.295 | 0.363 |
| rich | 2,001.0 - 19,671.0 | 0.256 | 0.295 | 0.363 |

Table 1: Average CLIP scores per income quartile for different visual encoders.

A similar trend can be observed when quartiles of the data distribution are used as income bins. In addition to CLIP ViT-B-32, we also conduct the same experiment for the following CLIP visual encoders: ViT-L-14 from OpenAI and ViT-G-14 (pre-trained on the LAION dataset). Table 1 shows the average CLIP scores aggregated across each quartile; the lowest income quartile has the lowest average CLIP score among all the other quartiles for the three CLIP model versions.

**What other factors affect CLIP performance?** A closer inspection of the images reveals that in addition to income, there are other factors that affect CLIP's performance such as diversity in topic appearance and topic subjectivity.

Figure 2 shows a qualitative analysis of five random topics across income levels. We observe that **diversity in topic appearance** is more predominant in lower-income households, probably due to a need to improvise (e.g."newspaper" or "grass") and less standardization across object appearances.

The **subjective topics** (e.g., "idols", "things I wish I had", "most loved item", "next big purchase") are also problematic for models to classify correctly as they are subject to different interpretations based on the individual's background and personal thoughts (e.g., one person's "next big purchase" is a "couch", while another's is a "cow" – Figure 8 in Appendix). Therefore, we chose to follow Ramaswamy et al. (2023) and not include these topics in our CLIP performance analysis.

Note that the purpose of Figure 2 is not to highlight CLIP's failures, since it would be expected to perform poorly on these examples. Instead, we emphasize the existence of diversity in topic appearance and subjective topics as challenges to vision-language models and address how to mitigate them in Section 5.

[$]www.worldbank.org

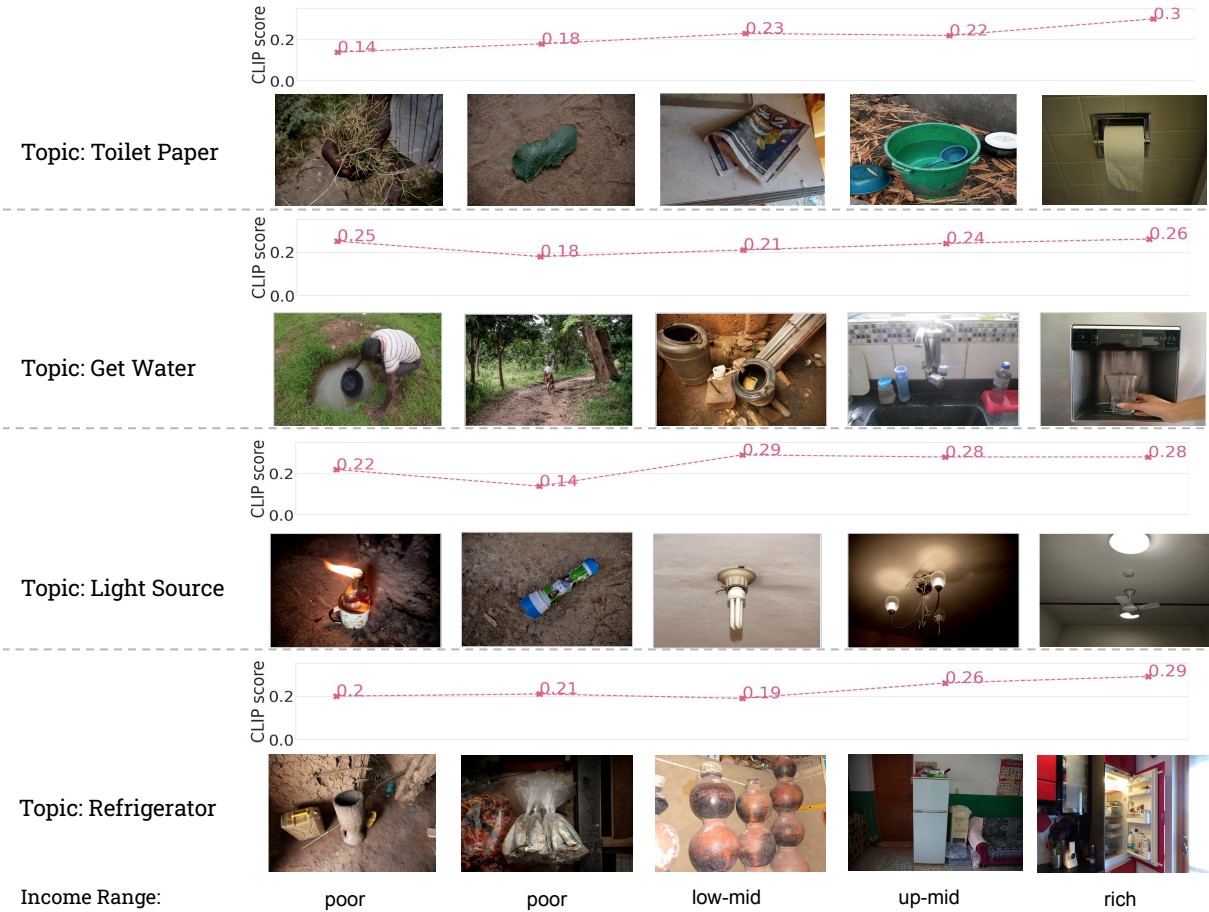

Figure 2: Qualitative analysis showing the data diversity across different income quartiles on five random topics: "toilet paper", "get water", "light source", "refrigerator". The CLIP performance on the same topic is influenced by the **remarkably diverse appearance** of entities from the *same* topic, which often correlates with income. Our analysis draws attention to how diverse objects and actions appear in our everyday lives and calls for future work to consider this when building models and datasets. Best viewed in color.

**When does CLIP perform surprisingly well for low-income images?** When inspecting low-income images with high CLIP scores, we find potentially problematic reasons for CLIP's high performance.

Specifically, because CLIP performs like a "bag of words" model ([Thrush et al., 2022](); [Castro et al., 2023]()), it has high performance when the image contains explicit visual representations of at least one word in the topic (e.g., "get water" in Figure 2 has a relatively high CLIP score if the "water" is visible in the image as a "lake" or a "puddle").

One approach to mitigate the occurrence of this problem involves the use of tools that provide a framework for inspecting images for possible confounders that might affect the performance of a model ([Wang et al., 2020]()).

**RQ2. How many relevant images are retrieved by CLIP across countries and income levels?**

This section provides a more fine-grained CLIP performance evaluation, where we measure how many relevant images are retrieved by CLIP for each topic, country, and income level.

Specifically, we compute and analyze the *recall score* across topics, income levels, and countries. The recall score is computed as follows. First, for each topic, we compute the CLIP similarity score between all the images in the dataset and a given topic. Next, we select the images with the top $N$ scores, where $N$ is the number of ground truth images corresponding to the topic $T$. We then measure the percentage of correct images (i.e., representing topic $T$) among these $N$ images. [$]

**Recall scores across income levels.** We show the CLIP recall over all images, across different income quartiles in Figure 3. The true positives

---

[$]In Information Retrieval, this is referred to as R-Precision. It is the cross point where precision and recall are equal.

($TP$) represent the images that were correctly "recognized", while the false negatives ($FN$) represent the images left out of the top $N$ scored images or "forgotten" images. As seen in Figure 3,

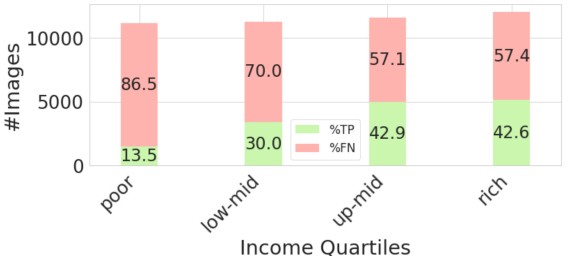

Figure 3: CLIP Recall over all images: percentage of true-positive or "recognized" images and false-negative or "forgotten" images for each income quartile. Increasingly more images are forgotten in the lower-income bins, with $86.5\%$ forgotten images in the *poor* quartile.

CLIP has unequal performance in retrieving images across different income groups and images collected from poor households are less likely to be retrieved correctly. We also find that out of the 199 topics that have image contributions from all four income groups, 137 (68.8%) topics show a similar trend in unequal performance as displayed in Figure 3 (see Appendix Table 2 for more details).

**Topics with highest recall deviations across income levels.** Ideally, CLIP's performance for a given topic should be relatively similar across all income levels. We calculate the variance between recall scores across income levels for each topic: Figure 4 displays the top ten topics with the highest recall variance between income levels.

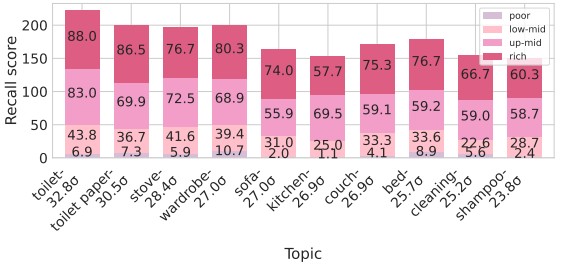

Figure 4: Top 10 topics with the largest standard deviation over the CLIP recall scores per income quartiles; recall varies the most across income on these topics.

As shown in the figure, topics like "toilet", "toilet paper", and "wardrobe" that have high recall scores for the rich income level ($> 80$) also have extremely low recall scores for the poor income level ($< 15$). This is concerning because CLIP

appears to perform well for a topic, while further analysis reveals that it has a dismal performance on data from the lowest income group, as our findings show (see more examples in Appendix Table 2).

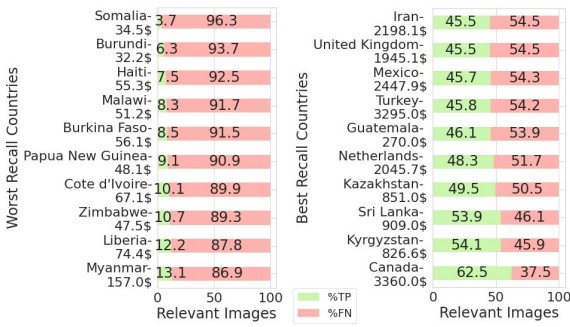

Figure 5: Countries with high and low recall scores: 7/10 countries with the worst recall have low average incomes and are from *Africa*. The countries with the best recall scores have high average incomes and are from *America, Europe, Asia*. Countries with low recall also have low income, while most countries (apart from *Guatemala*) with high recall have a high income.

**Disparate recall across countries.** We compute the recall over all images, across countries. Figure 5 compares the top ten countries with the best and worst ratio of true-positives to false-negatives. The countries with the best recall have true-positive rates up to $40\%$ higher than those with the worst recall. Note that most countries in the worst recall plot have low average incomes and are in *Africa*. The average income per country is computed as the mean income of the dataset households in each country, which follows the real-world trend[§].

**RQ3. What is the diversity of topic appearances across countries and income levels?** Analyzing the findings from the first two RQs, we observe that topics from lower-income households tend to have more diverse appearances. This is important because, intuitively, topics with diverse appearances are more prone to vision-language model performance variations. For instance, the topic "toilet paper" appears in Figure 4 among those with the most variance in recall across income levels (poor: 7.3; low-mid: 36.7; up-mid: 69.9; rich: 86.5), and with at least five diverse appearances ("grass", "leaf", "newspaper", "water" and "toilet paper") in Figure 2.

[§]https://datatopics.worldbank.org/world-development-indicators/the-world-by-income-and-region.html

To automatically discover which topics have diverse appearances and identify the different forms that the images belonging to one topic could take – e.g., "toilet paper" taking the form of "grass", "leaf" or "newspaper"– we find inspiration from *word analogies* (Bolukbasi et al., 2016), such as "man is *to* programmer as woman is *to* homemaker" and develop *topic analogies* for the visual representations of topics across countries and income levels: e.g., "toilet is *in* poor Nigerian households as backyard is *in* rich Chinese households.

**Visual Similarity.** First, we visually represent a topic for a given country and income level – *(topic, country, income)* – as the average of all the CLIP visual representations of the corresponding images. Next, we compute the visual similarity between *(topic, country, income)* tuples as the cosine similarity between their visual representations.

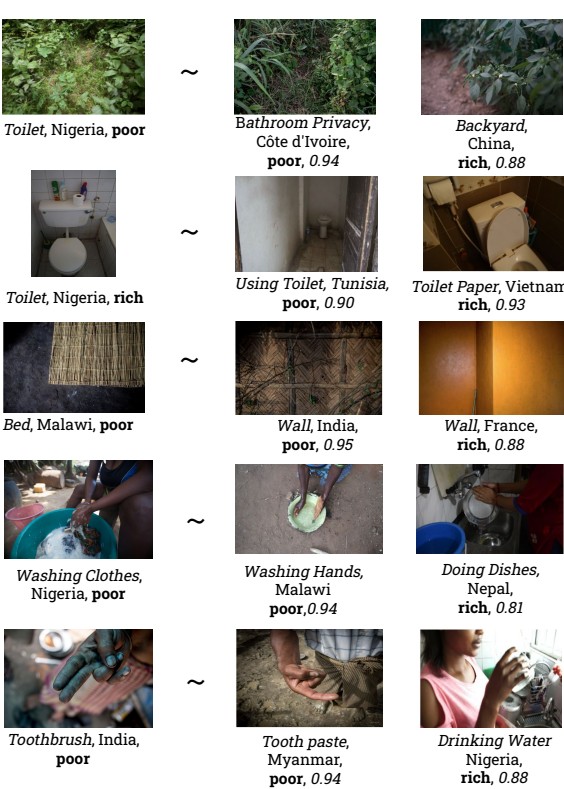

Figure 6: For a given query-tuple (topic, country, income) on the left (e.g., "toilet paper in poor Nigeria"), we show the most visually similar tuples on the right. Best viewed in color.

In Figure 6, we show examples of visual similarities between a query-tuple and the returned tuples. We find compelling results, such as:

1. Toilet is in poor Nigerian households as Backyard is in rich Chinese households.

2. Bed is in poor Malawian households as Wall is in rich French households.

3. Washing Clothes is in poor Nigerian households as Doing Dishes is in rich Nepalese households.

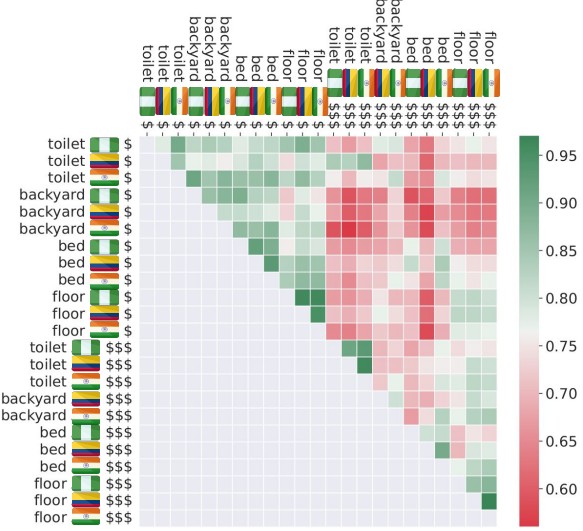

Figure 7: Heatmap of the visual similarity scores between images from different income levels (*poor $, rich $$$*), from three countries on different continents (*Nigeria* 🇳🇬 , *Colombia* 🇨🇴 , *India* 🇮🇳 ), for four common topics (*toilet, backyard, bed, floor*). Best viewed in color.

In Figure 7, we compute a heatmap of the visual similarity scores between images from different income levels, from three countries on different continents, for four common topics. Overall, we see how topics from poor households tend to be more similar to each other, in the green/ high similarity cluster, and not similar to topics from rich households, in the red/ low similarity cluster.

We quantify appearance diversity for each topic across rich and poor income levels by counting how many of the visually similar topics are *actually* similar, i.e., textually similar – e.g., in Figure 6 "bed" and "wall" are visually similar, but not textually similar. We compute the textual similarity as the cosine similarity between the sentence embeddings (Reimers and Gurevych, 2019) of the query topic and the returned top ten most visually similar topics. We count the topics as actually similar if their textual similarity is greater than a threshold, 0.7. We find that topics from rich households have 33% actually dissimilar topics, while topics from poor households have 47%, demonstrating

that topic diversity is higher in poor households than in rich households.

## 5 Lessons Learned and Actionable Steps

Our analyses yield several insights into the current state of vision-language models. We highlight lessons learned and propose actionable steps to ensure future work builds more equitable models and datasets.

**Invest effort to understand the extent of the digital divide in vision-language performance.** Four years after De Vries et al. (2019) drew attention to the performance disparity across income levels for object recognition models, we show that a popular vision-language model (CLIP) also yields poor performance on data from lower-income households. This new finding, together with the scarcity of research in this area, shows that the AI research community is not paying enough attention to poor socio-economic groups worldwide. The consequences can be devastating since, if neglected, AI will exponentially increase the already existing "digital divide".

**Define evaluation metrics that represent everyone.** We call for attention and scrutiny of broad performance metrics that give the illusion of high accuracy in AI models while performance on minority groups remains dangerously inadequate. One potential solution is to use the geo-location information of images when available to evaluate models by location and income, as income can often be deduced from the location.

**Match diversity standards to model purpose.** Diversity standards for AI models and datasets must match the intended users. For example, an AI model built as a foundation baseline for various downstream tasks should be held to the highest standards regarding what training data is used, as it should be able to represent everyone equally.

**Document training data.** The origins of a model's training dataset provide valuable information to prospective users and developers regarding the limitations and best use cases for a model. Model creators are responsible for documenting the data the model was trained on and any known biases it may have (Mitchell et al., 2019; Gebru et al., 2021; Wang and Russakovsky, 2023).

**Invest in geo-diverse datasets.** In as much as collecting geographically diverse datasets can be expensive compared to web scraping methods, they have been proven to improve overall model performance and mitigate bias (Ramaswamy et al., 2023; Kirillov et al., 2023). In contrast, datasets created using web scraping exclude data from households with limited/no access to the Internet, which reflects a staggering 37% of the world's population.[$] A solution is to supplement these datasets through crowd-sourcing to allow for broader reach. Similarly, as shown in Figure 5, more attention should be given to data collection from underrepresented countries.

**Annotate diversity and subjectivity in datasets.** For actions or entities that can be represented by remarkably different items that perform the same functionality, as shown in Figure 2, it is essential to reconcile these differences and inclusively categorize them. Furthermore, as highlighted by Ignat et al. (2019), people often use various names for the same action or object, and having just one label limits the diversity of our vocabulary. Ramaswamy et al. (2023) also found this challenging when collecting data: "stove" was originally underrepresented until the definition of "stove" was clarified, as workers did not always consider their cooking appliances to be a "stove". One possible solution might involve the development of *flexible labels*, also considering the data provider.

Furthermore, Chan et al. (2021) note that *participation in data collection does not always equate to representation*, as the research labs often dictate how data is labeled and categorized; models trained on diverse datasets still make distortions through a Western lens and not truly reflect the culture of the represented people. To ensure that data from across all income groups and countries are labeled to reflect the people's authentic culture, representatives from these groups should also participate in the labeling process.

## 6 Conclusion

In this paper, we evaluated the performance of a state-of-the-art vision-language model (CLIP) on household images across different income groups, and different countries as found in the DollarStreet dataset. The results of our analyses demonstrated the existence of performance inequality and showed that data from lower-income groups are more likely to be represented inaccurately.

---

[$] https://www.un.org/en/delegate/itu-29-billion-people-still-offline

To mitigate these issues, we analyzed and quantified the visual appearance diversity in topics – a current challenge for vision-language models – and determined that there is a higher topic visual appearance diversity in low-income images than higher income images. Furthermore, building on our findings, we shared six actionable steps for improving inclusion in AI development.

We hope our work will contribute to a better understanding of the limitations of current vision-language models and provide important directions to guide future research toward "democratizing" vision-language models and datasets.

## Limitations

Out of all the publicly available image datasets, Dollar Street is the most diverse dataset that contains income information. However, some limitations exist in terms of country and income representation, since Dollar Street does not capture all representative households worldwide.

The dataset contains data from 63 countries, representing four continents (Africa, America, Asia, and Europe). Therefore, our analyses do not account for 121 countries in the aforementioned continents and the entire regions of Australia and Antarctica. Some countries have sparse contributions from only one or two households. For example, Canada has only 45 images, all contributed by one household. Since most European households in Dollar Street belong to the upper-middle and rich income group and about half of the African households belong to the poor and lower-middle income group, our study does not capture the complete representation of the different economic groups in those countries.

## Acknowledgement

We thank the anonymous reviewers for their constructive feedback. We are also grateful to Artem Abzaliev, Ashkan Kazemi, and Longju Bai for proofreading the manuscript and providing several suggestions that helped shape the research questions. Lastly, we thank the members of the Language and Information Technologies lab at the University of Michigan for the insightful discussions during the early stage of the project. This project was partially funded by a grant from the Templeton Foundation (#62256) and a grant from the Department of State (#STC10023GR0014). Any opinions, findings, and conclusions or recommendations expressed in this material are those of the authors and do not necessarily reflect the views of the Templeton Foundation or the Department of State.

## Ethical Considerations

Several ML models have been used to estimate the income bracket of neighborhoods. Since Dollar Street provides image data along with household income values, there is the potential for ML models to be trained on the dataset to predict income (Acharya et al., 2017; Gebru et al., 2017; Yeh et al., 2020a; Machicao et al., 2022).

We note that our analyses suggest that vision-language models could indirectly identify which images belong to lower-income groups. However, our work does not contribute to or encourage the disingenuous use of AI for tasks that lead to discrimination or unethical applications. Instead, we create awareness of AI performance disparity across income groups and call the research community to action toward mitigating them.

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

# A    Subjective Topics

The 19 subjective topics that we remove: "favorite home decorations", "favourite item in kitchen", "favourite sports clubs", "how the most loved item is used", "icons", "idols", "latest furniture bought", "looking over the shoulder", "most loved item", "most loved toy", "most played songs on the radio", "music idol", "next big thing you are planning to buy", "playing with most loved toy", "thing I dream about having", "things I wish I had", "using most loved item", "youth culture", "what I wish I could buy".

# B    CLIP performance - other analyses

Figure 10 displays topics with trends that deviate from the plot shown in Figure 3.

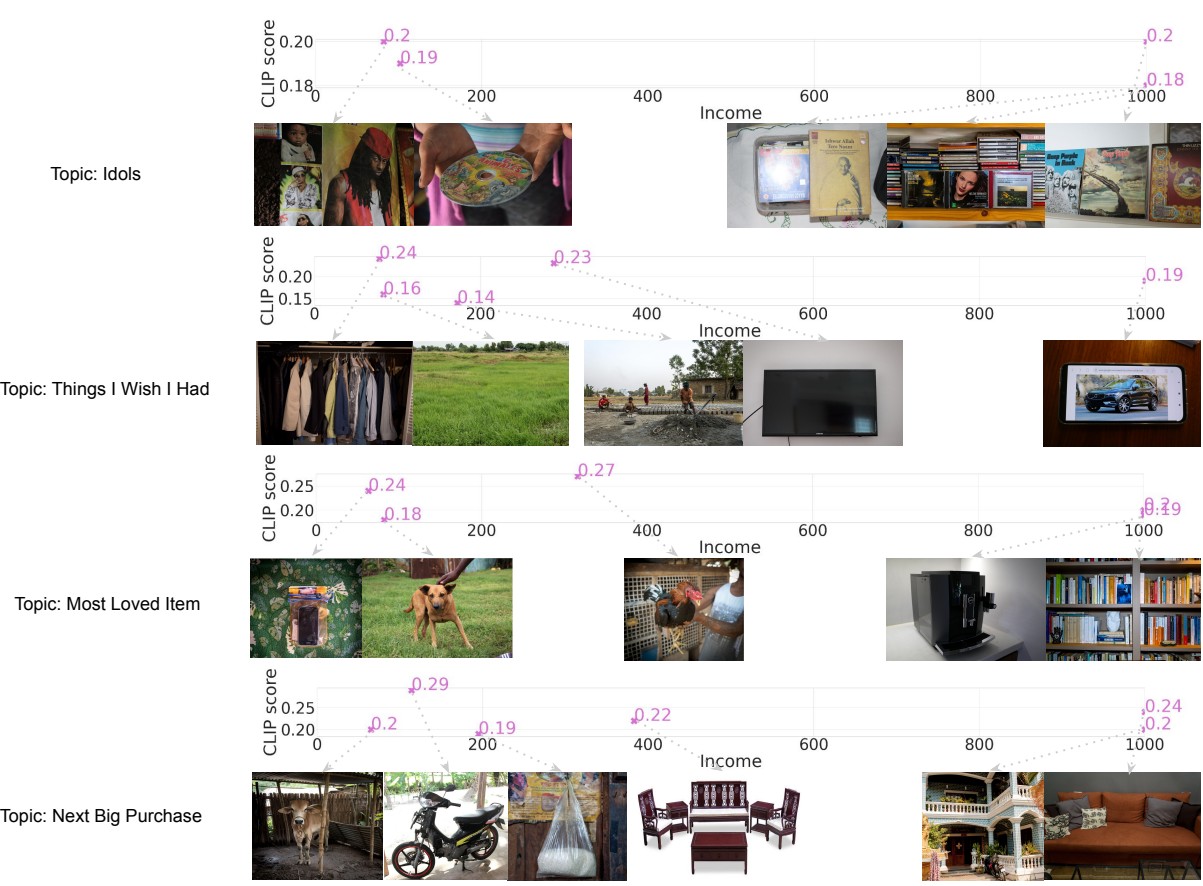

Figure 8: Qualitative analysis showing the CLIP performance on **subjective data** across different incomes on four random topics: "idols", "things I wish I had", "most loved item", "next big purchase".

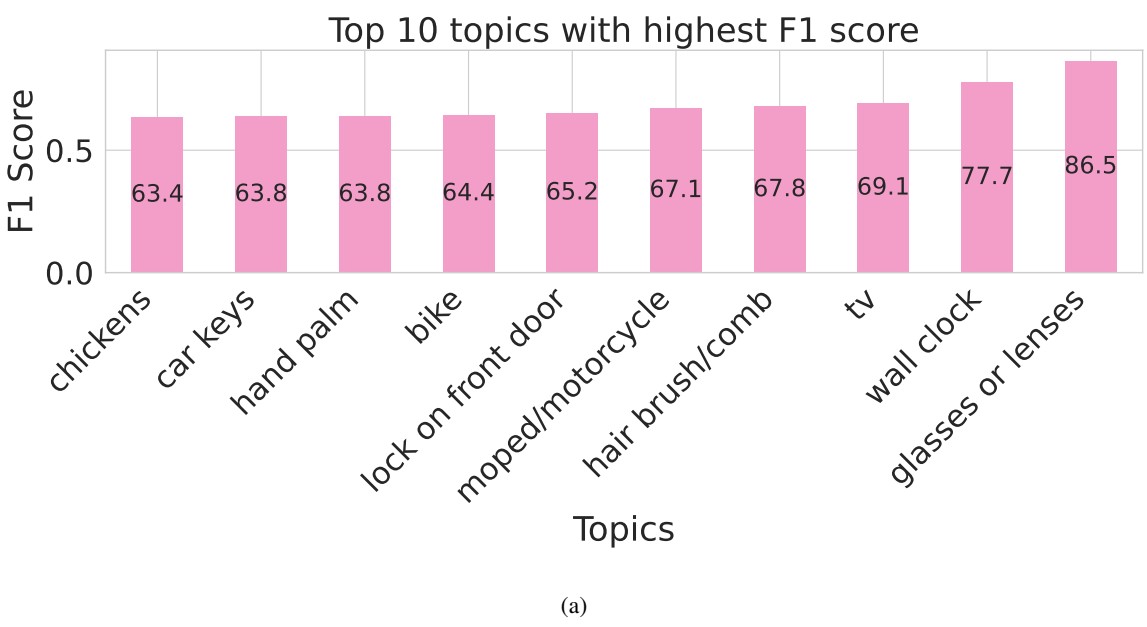

(a)

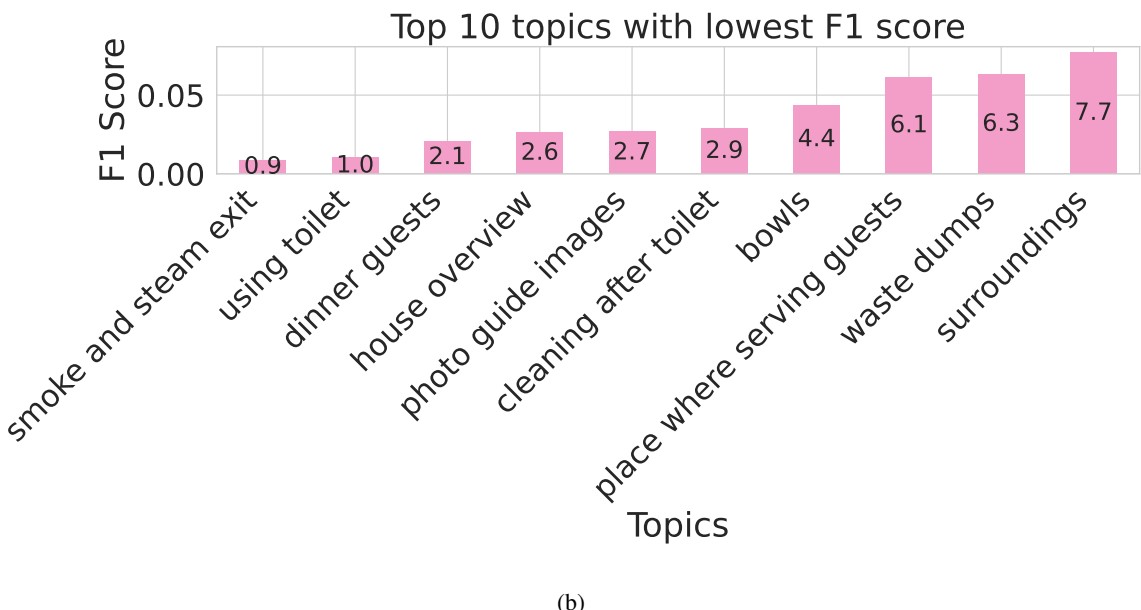

(b)

Figure 9: Top 10 topics with highest (a) and lowest (b) F1 scores.

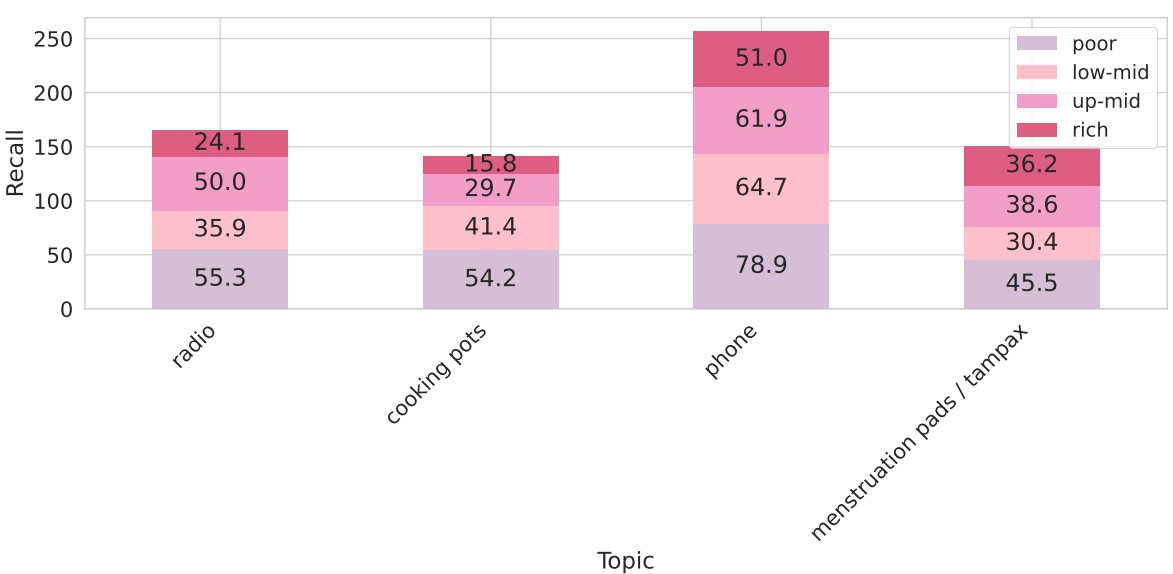

Figure 10: Topics where recall for lowest income level is higher than recall for other income levels.

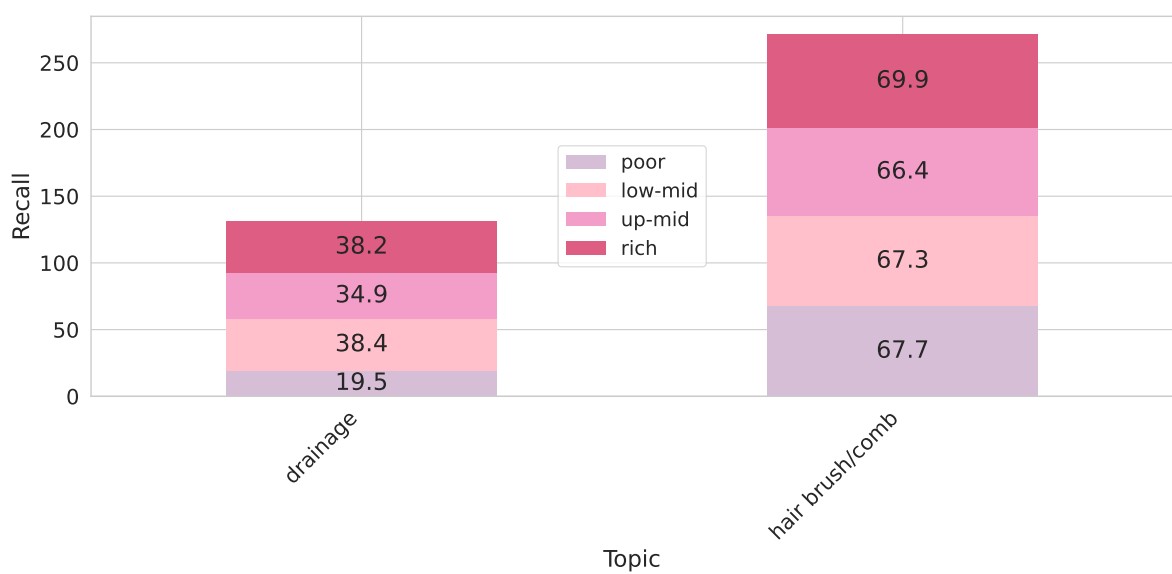

Figure 11: Topics where recall for is relatively high (hairbrush) and relatively low (drainage) across all income levels

| Topic | Poor | Low-mid | Up-mid | Rich | Topic | Poor | Low-mid | Up-mid | Rich |
|---|---|---|---|---|---|---|---|---|---|
| adding spices to food while cooking | 0.0 | 0.0 | 10.0 | 14.7 | soap for hands and body | 11.0 | 20.9 | 42.1 | 37.7 |
| spices | 20.6 | 35.3 | 53.6 | 56.5 | shampoo | 2.4 | 28.7 | 57.6 | 60.3 |
| answering the phone | 0.0 | 0.0 | 10.0 | 15.4 | kitchen sink | 0.0 | 24.6 | 72.3 | 65.2 |
| jewelry | 10.0 | 25.0 | 46.4 | 52.5 | dishwasher | 0.0 | 0.0 | 23.1 | 33.3 |
| armchair | 24.6 | 41.1 | 55.6 | 66.7 | doing dishes | 0.0 | 7.3 | 40.6 | 44.2 |
| couch | 4.1 | 33.3 | 59.1 | 75.3 | social drink | 20.7 | 26.2 | 26.2 | 29.0 |
| sofa | 2.0 | 31.0 | 55.9 | 74.0 | drying | 7.7 | 33.3 | 42.0 | 40.3 |
| pet | 10.0 | 16.9 | 30.2 | 28.6 | washing clothes/cleaning | 5.6 | 22.6 | 59.0 | 65.7 |
| place where eating dinner | 7.1 | 23.4 | 44.9 | 47.6 | earings | 41.7 | 50.0 | 58.5 | 60.0 |
| sitting area | 0.0 | 19.0 | 33.8 | 35.7 | necklaces | 40.0 | 42.9 | 63.6 | 68.4 |
| backyard | 2.6 | 17.5 | 42.3 | 50.0 | piercings | 11.1 | 40.9 | 52.9 | 65.0 |
| drinking water | 10.1 | 32.2 | 40.9 | 44.1 | eating | 0.0 | 7.4 | 37.5 | 27.3 |
| street view | 28.8 | 62.1 | 64.1 | 71.6 | family eating | 22.2 | 44.0 | 50.0 | 53.8 |
| toilet | 6.9 | 43.8 | 83.0 | 88.0 | everyday shoes | 18.5 | 42.7 | 62.0 | 64.0 |
| bathroom privacy | 0.0 | 4.3 | 21.3 | 33.9 | nicest shoes | 27.4 | 39.1 | 61.3 | 68.3 |
| trash/waste | 10.2 | 37.2 | 45.6 | 57.3 | family | 10.7 | 40.7 | 50.9 | 42.6 |
| kitchen | 1.1 | 25.0 | 69.5 | 59.2 | family snapshots | 4.8 | 17.9 | 26.1 | 27.0 |
| bathroom/toilet | 1.0 | 31.0 | 60.7 | 45.3 | wall decoration | 4.9 | 21.8 | 33.7 | 38.7 |
| hand washing | 12.3 | 24.4 | 40.3 | 71.9 | wedding photos | 50.0 | 13.3 | 61.5 | 55.6 |
| shower | 0.0 | 18.8 | 56.6 | 64.9 | teeth | 41.7 | 49.4 | 76.7 | 86.4 |
| bed | 8.9 | 33.6 | 60.2 | 76.7 | refrigerator | 0.0 | 38.4 | 57.3 | 48.3 |
| bed kids | 3.8 | 14.7 | 41.4 | 33.3 | rug | 19.0 | 51.9 | 72.9 | 66.7 |
| bedroom | 4.2 | 27.5 | 45.8 | 59.2 | freezer | 20.0 | 35.1 | 38.7 | 50.0 |
| guest bed | 0.0 | 8.7 | 20.9 | 36.4 | front door | 17.5 | 38.9 | 68.5 | 53.8 |
| toys | 17.6 | 25.8 | 37.2 | 34.5 | lock on front door | 41.5 | 69.6 | 78.1 | 71.6 |
| children room | 1.6 | 19.2 | 38.0 | 39.4 | moped/motorcycle | 66.7 | 48.0 | 71.4 | 85.0 |
| cleaning equipment | 1.1 | 9.8 | 20.0 | 21.1 | opening the front door | 0.0 | 8.0 | 34.8 | 22.9 |
| play area | 7.2 | 14.6 | 40.3 | 44.9 | fruit trees | 25.7 | 52.0 | 54.2 | 56.2 |
| living room | 0.0 | 19.7 | 53.4 | 61.7 | fruits and vegetables | 6.2 | 27.7 | 53.3 | 54.9 |
| bike | 48.1 | 63.2 | 72.3 | 67.6 | grains | 28.2 | 29.4 | 30.0 | 47.8 |
| books | 21.9 | 42.5 | 65.5 | 78.2 | vegetables | 20.3 | 41.3 | 62.8 | 60.6 |
| paper | 3.8 | 18.2 | 24.0 | 29.1 | shaving | 0.0 | 15.1 | 29.2 | 30.8 |
| dish racks | 2.2 | 25.3 | 48.2 | 58.1 | hallway | 0.0 | 25.0 | 33.3 | 37.5 |
| plates | 17.0 | 32.2 | 62.5 | 62.7 | hand back | 2.7 | 10.8 | 21.0 | 13.5 |
| brushing hair | 0.0 | 4.5 | 11.1 | 25.0 | hand open to closed | 0.0 | 0.0 | 44.4 | 29.4 |
| brushing teeth | 4.6 | 26.1 | 35.7 | 50.0 | washing hands | 0.0 | 24.4 | 77.4 | 81.4 |
| toothbrush | 17.0 | 37.3 | 65.5 | 64.4 | playing | 7.7 | 7.7 | 28.6 | 13.8 |
| car | 0.0 | 14.3 | 44.4 | 36.5 | light source in kitchen | 0.0 | 19.5 | 48.1 | 41.8 |
| frontdoor keys | 0.0 | 13.8 | 27.3 | 22.2 | lightsources by bed | 0.0 | 26.7 | 31.0 | 28.6 |
| wheel barrow | 22.2 | 46.7 | 57.1 | 57.1 | reading light | 0.0 | 6.7 | 20.7 | 20.6 |
| parking lot | 0.0 | 10.7 | 20.0 | 45.8 | tools | 5.8 | 30.5 | 40.6 | 52.4 |
| get water | 2.0 | 8.3 | 21.9 | 13.7 | plate of food | 11.0 | 46.0 | 47.4 | 46.2 |
| ceiling | 9.9 | 25.3 | 59.0 | 63.0 | wall inside | 16.0 | 28.0 | 39.3 | 38.6 |
| light source in livingroom | 1.3 | 21.8 | 44.3 | 44.0 | pet foods | 0.0 | 14.3 | 68.8 | 59.3 |
| light sources | 10.2 | 19.0 | 35.6 | 32.7 | plugging into and out of power outlet | 0.0 | 33.3 | 51.5 | 40.0 |
| wall clock | 36.0 | 74.1 | 89.7 | 89.4 | power outlet | 14.6 | 46.4 | 69.7 | 64.5 |
| chickens | 60.5 | 63.6 | 75.0 | 66.7 | pouring drinking water | 1.7 | 8.6 | 26.3 | 21.1 |
| meat or fish | 25.0 | 20.0 | 42.9 | 45.0 | pouring water | 1.7 | 10.5 | 25.9 | 32.1 |
| chopping ingredients | 0.0 | 27.8 | 40.0 | 43.5 | switch on/off | 5.8 | 36.5 | 50.0 | 62.9 |
| chopping food | 3.7 | 26.9 | 50.0 | 52.9 | listening to the radio | 0.0 | 8.3 | 15.4 | 11.1 |
| toilet paper | 7.3 | 36.7 | 69.9 | 86.5 | reading | 14.3 | 37.0 | 70.6 | 72.2 |
| cleaning floors | 0.0 | 14.7 | 44.2 | 47.4 | reading a book | 0.0 | 41.9 | 70.3 | 76.3 |
| floor | 14.4 | 38.5 | 60.6 | 56.6 | salt | 19.0 | 31.5 | 40.2 | 45.1 |
| washing detergent | 0.0 | 9.8 | 21.5 | 24.6 | taking a teaspoon of salt | 0.0 | 13.0 | 33.3 | 26.3 |
| closing the front door | 0.0 | 5.0 | 28.6 | 29.0 | shoes | 1.4 | 25.5 | 43.2 | 62.7 |
| wardrobe | 10.7 | 39.4 | 68.9 | 80.3 | sleeping | 0.0 | 50.0 | 61.5 | 52.2 |
| coats and jackets | 0.0 | 14.3 | 41.7 | 50.0 | drinking social drink | 15.8 | 14.3 | 17.4 | 17.1 |
| work area | 0.0 | 13.3 | 44.2 | 60.9 | storage room | 2.6 | 13.0 | 25.0 | 40.5 |
| cooking | 13.8 | 19.6 | 39.1 | 32.6 | street detail | 4.9 | 17.5 | 29.0 | 43.5 |
| stove/hob | 5.9 | 41.6 | 72.5 | 76.7 | source of light | 0.0 | 17.6 | 34.5 | 30.6 |
| knifes | 31.0 | 51.1 | 77.6 | 70.7 | worship places | 0.0 | 11.8 | 37.5 | 20.6 |
| preparing food | 6.9 | 6.5 | 28.6 | 23.3 | toothpaste on toothbrush | 0.0 | 0.0 | 25.0 | 13.6 |
| cosmetics | 0.0 | 43.8 | 61.1 | 55.3 | seeing the back of book | 0.0 | 0.0 | 27.3 | 13.6 |
| cups/mugs/glasses | 29.5 | 46.3 | 64.7 | 74.6 | turning heater on | 0.0 | 0.0 | 44.4 | 16.7 |
| tooth paste | 21.0 | 52.5 | 66.4 | 59.8 | walking towards front door | 0.0 | 4.8 | 13.0 | 26.5 |
| diapers (or baby-pants) | 28.6 | 65.2 | 70.6 | 87.5 | water outlet | 1.2 | 19.2 | 30.1 | 39.7 |
| dish washing brush/cloth | 2.0 | 5.8 | 19.3 | 26.7 | worshipping | 0.0 | 4.0 | 15.4 | 36.4 |
| dish washing soap | 0.0 | 11.9 | 34.9 | 50.0 | wall | 9.0 | 18.8 | 33.3 | 36.0 |
| hand palm | 48.3 | 58.2 | 76.6 | 86.0 | | | | | |

Table 2: Topics that have a similar plot as Figure 3 and Recall values for Poor, Low-mid, Up-mid, Rich income levels: Where recall for rich and up-mid is higher than recall for poor and low-mid.

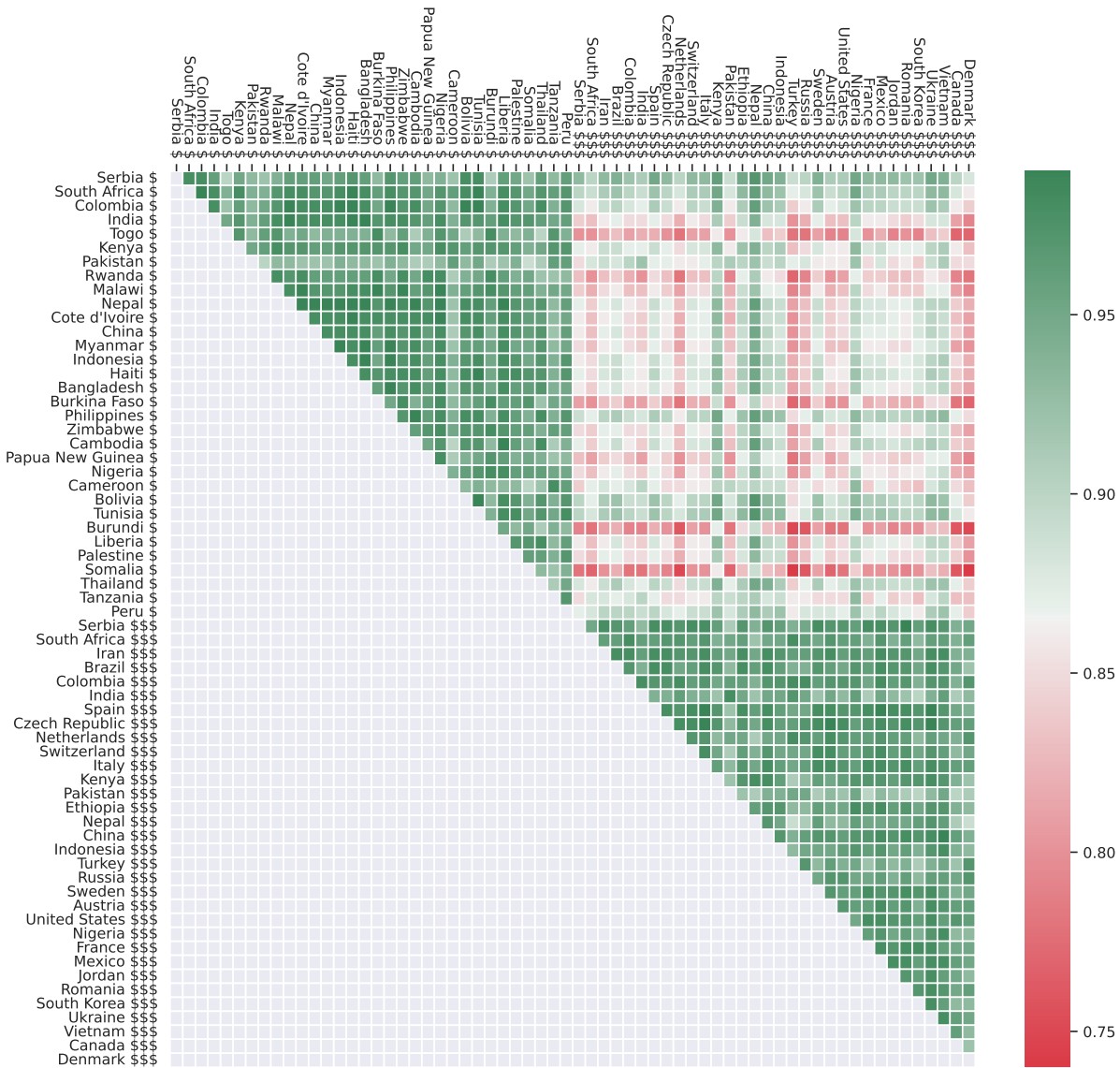

Figure 12: Heatmap of the visual similarity scores between images from different income levels (*poor $, rich $$$*), from sixty-four countries on different continents across all topics. Best viewed in color.