# OpenReview forum: "Bridging the Digital Divide: Performance Variation across Socio-Economic Factors in Vision-Language Models"
_EMNLP/2023/Conference — EMNLP 2023 Main_

### Official Review · Reviewer_bJzE · 2023-07-31

**Soundness:** 4

**Excitement:**

4: Strong: This paper deepens the understanding of some phenomenon or lowers the barriers to an existing research direction.

**Paper Topic And Main Contributions:**

This paper uses an existing dataset of geo-coded images with category labels to investigate the economic (and geographic) performance distribution of a population Vision-Language model. The main findings are a gap between the poorest households and all other socio-economic levels. The analysis takes into account the social structure of various image topics (c.f. Figure 2) and overall provides a helpful discussion of the sources of this disparity.

**Reasons To Accept:**

This paper is a clearly written investigation of an important topic (economic / geographic bias in pre-trained models). The results show a meaningful difference in performance between the lowest income groups and other groups. And the discussion helps us situate and interpret that gap. Overall, the paper makes a convincing case that these sorts of geographic/demographic approaches to evaluation are important for ensuring that models generalize well.

**Reasons To Reject:**

This a strong paper and there are no substantial reasons to reject it.

**Reproducibility:**

4: Could mostly reproduce the results, but there may be some variation because of sample variance or minor variations in their interpretation of the protocol or method.

**Reviewer Confidence:**

3: Pretty sure, but there's a chance I missed something. Although I have a good feel for this area in general, I did not carefully check the paper's details, e.g., the math, experimental design, or novelty.

---

> ### Author Rebuttal · Authors · 2023-08-28
>
> Thank you so much for your review, we appreciate your kind comments. We are very glad to hear that you found our paper to be clearly written and the findings to be meaningful.

---

### Official Review · Reviewer_hF5y · 2023-08-04

**Soundness:** 4

**Excitement:**

5: Transformative: This paper is likely to change its subfield or computational linguistics broadly. It should be considered for a best paper award. This paper changes the current understanding of some phenomenon, shows a widely held practice to be erroneous in someway, enables a promising direction of research for a (broad or narrow) topic, or creates an exciting new technique.

**Missing References:**

This paper may be relevant from a geographic analysis standpoint, though it only examines geographic differences within the US. I imagine the disparate differences would hold across income variations within a country.

Lwowski, Brandon, Paul Rad, and Anthony Rios. "Measuring Geographic Performance Disparities of Offensive Language Classifiers." Proceedings of the 29th International Conference on Computational Linguistics. 2022.

**Paper Topic And Main Contributions:**

This paper investigates the digital divide in terms of model performance of OpenAI's CLIP image-text model. The paper shows that CLIP scores or correlated with income level, R-Precision differs substantially across income levels (i.e., image retrieval quality given a topic), and also observes differences across countries where Africa-related images are observed to have the worst performance. The fine-grained analysis observes that the differences in performance occur because of larger image variations in low-income locations/settings. Finally, the paper introduces important lessons learned, e.g., introducing geographically-diverse datasets and ensuring that the data diversity standards/performance of models should match their intended use (e.g.,  general purpose models should have the highest standard).

**Questions For The Authors:**

Question A: How would you expect model performance to change based on model size?

Question B: Would you expect multilingual variations of CLIP to have different findings?

**Reasons To Accept:**

Overall, this is a very well-written paper. The evaluation is important and novel. Moreover, the experiments are well done and showcase robust performance disparities across income levels.

**Reasons To Reject:**

(Minor) The model only explores CLIP, and more specifically, thy only use a single model variation of CLIP (ViT-B/32). Larger CLIP versions have better performance (ViT-L-14). Moreover, there are multilingual versions of CLIP that seem like a natural model to explore in these scenarios, even if all of the data is in English. Hence, there are natural unaddressed questions such as "How does model size impact performance disparities?" and "How does multilingual training impact performance disparities?" It is entirely possible that more accurate are more biased. Though, we can not know that without proper experiments.


REFERENCES:
Carlsson, Fredrik, et al. "Cross-lingual and multilingual clip." Proceedings of the Thirteenth Language Resources and Evaluation Conference. 2022.

Wang, Jialu, Yang Liu, and Xin Wang. "Assessing Multilingual Fairness in Pre-trained Multimodal Representations." Findings of the Association for Computational Linguistics: ACL 2022. 2022.

Dong, Xiaoyi, et al. "Clip itself is a strong fine-tuner: Achieving 85.7% and 88.0% top-1 accuracy with vit-b and vit-l on imagenet." arXiv preprint arXiv:2212.06138 (2022).

**Reproducibility:**

5: Could easily reproduce the results.

**Reviewer Confidence:**

4: Quite sure. I tried to check the important points carefully. It's unlikely, though conceivable, that I missed something that should affect my ratings.

---

> ### Author Rebuttal · Authors · 2023-08-28
>
> **Note of Thanks**
>  We are grateful for your positive comments on our paper, and we are excited to see that you find the paper to be very well-written and the evaluation important and novel.
>
> **RC: Reviewer's comment**
> **AR: Author's response**
>
> _**Reasons to Reject**_
>
> **RC** The model only explores CLIP, and more specifically, they only use a single model variation of CLIP (ViT-B/32). Larger CLIP versions have better performance (ViT-L-14). Moreover, there are multilingual versions of CLIP that seem like a natural model to explore in these scenarios, even if all of the data is in English. Hence, there are natural unaddressed questions such as "How does model size impact performance disparities?" and "How does multilingual training impact performance disparities?" It is entirely possible that more accurate are more biased. Though, we can not know that without proper experiments.
>
> **AR** _We performed subsequent analyses, which show that a similar performance variation trend also exists for larger CLIP models as discussed in our answers to your questions (below)._
>
> _**Questions for Authors**_
>
> **RC** Question A: How would you expect model performance to change based on model size?
>
>
>
> **AR** _Before conducting the experiments, we expect that larger CLIP models would show a similar performance variation as CLIP ViT-B-32 which we used in our paper. We conducted the RQ1 experiment, which measures average CLIP scores across income bins, for the following CLIP versions: ViT-L-14 from OpenAI and ViT-G-14 pre-trained on LAION, and we found that the trend of decreasing average CLIP scores from high-income to lower-income household images still holds as shown in the table below._
>
>
> | **Quartile name**  |**Income range**  | **Avg CLIP Score (ViT-B/32)**  | **Avg CLIP Score (ViT-L/14)**  | **Avg CLIP Score (ViT-G/14)** |
> |--------------------------|--------------------------|------------------------------------------|------------------------------------------|------------------------------------------|
> |poor                        | 26.9 - 195.0           |  0.233                                       | 0.259                                        | 0.321                                         |
> |low-mid                   | 195.4 - 685.0         | 0.250                                        | 0.282                                        | 0.350                                         |
> |up-mid                    | 694.0 - 1,998.0      | 0.257                                        | 0.295                                        | 0.363                                         |
> |rich                         | 2,001.0 - 19,671.0 | 0.256                                        | 0.295                                        | 0.363                                         |
>
>
> _To determine which models perform better on low-income data, we also intend to calculate R-precision scores following the RQ2 experiments for each model and include these results in the camera-ready version._
>
> **RC** Question B: Would you expect multilingual variations of CLIP to have different findings?
>
>
> **AR**
> _This is a very interesting suggestion, thank you. We expect roughly similar performance but we plan to explore this in future work._
>
> _**Missing References**_
>
> **AR** We acknowledge that these papers are relevant to our work and we will add them to the camera-ready version.

---

### Official Review · Reviewer_Vvi2 · 2023-08-05

**Soundness:** 3

**Excitement:**

3: Ambivalent: It has merits (e.g., it reports state-of-the-art results, the idea is nice), but there are key weaknesses (e.g., it describes incremental work), and it can significantly benefit from another round of revision. However, I won't object to accepting it if my co-reviewers champion it.

**Missing References:**

Visually Grounded Reasoning across Languages and Cultures. Liu et al., 2021
Broaden the Vision: Geo-Diverse Visual Commonsense Reasoning. Yin et al., 2021
GIVL: Improving Geographical Inclusivity of Vision-Language Models with Pre-Training Methods. Yin et al., 2023

**Paper Topic And Main Contributions:**

The paper studies the gap in the evaluation of V&L models, specifically their performance on underrepresented groups. The researchers assess CLIP using a diverse dataset of household images linked to income levels, DollarStreet. The findings reveal performance disparities between different income groups, with consistently lower performance for poorer households compared to wealthier ones across topics and countries. The study emphasizes the need to address these inequalities and proposes actionable strategies for creating AI models that are more inclusive across economic levels.

**Questions For The Authors:**

Why not just use object classification accuracy but use the simple CLIP scores?

For visual similarity section, is there any way to extend the analysis to larger-scale?

**Reasons To Accept:**

Great motivation and interesting research question: Studying the correlation between socio-economic factors and VL model performance is very important to let people aware of the potential bias and the severity of performance disparity for the applications. I’m impressed with Section 5 which revisits the problem and provide inspiring future steps.

Comprehensive investigation on the quality of generated instruction data: I'm glad to see that the authors utilize different metrics to evaluate the performance disparity across different income levels and countries. The comprehensive evaluation helps us better understand the effect of proposed methods. I’m especially impressed by multi-grained experiments that comprehensively support the authors’ arguments.

**Reasons To Reject:**

Evaluation is not rigorous: When using CLIP score to identify if the text correctly describes the object in the image, it is very important to *NORMALIZE* the score. For example, we first need to calculate the score between the image and all the object categories. And then we normalize all the scores and pick up the maximal one. We can’t directly calculate the score, because even if “toilet paper” has high matching score with the image, there might be another object name that has higher score. The evaluation may hurt the soundness of the method a little bit.

May lack more interesting findings: Although the authors provide very comprehensive analysis and in-depth insights, most of them are very similar to what previous works did. For example, in Liu et al., 2021 and Yin et al., 2021, the authors already find that there’s a large performance disparity across Western and non-Western countries. Yin et al., 2023 also quantify the performance gap on Dollar Street. Though it is not going down to the salary level, the basic observation is similar. What I expect is that, besides claiming that the models are biased, please also try to dig out the reason and the ways we can improve. That can make the paper more convincing and attract more people to follow and devote to make the models better.
Visually Grounded Reasoning across Languages and Cultures. Liu et al., 2021
Broaden the Vision: Geo-Diverse Visual Commonsense Reasoning. Yin et al., 2021
GIVL: Improving Geographical Inclusivity of Vision-Language Models with Pre-Training Methods. Yin et al., 2023

**Reproducibility:**

4: Could mostly reproduce the results, but there may be some variation because of sample variance or minor variations in their interpretation of the protocol or method.

**Reviewer Confidence:**

3: Pretty sure, but there's a chance I missed something. Although I have a good feel for this area in general, I did not carefully check the paper's details, e.g., the math, experimental design, or novelty.

---

> ### Author Rebuttal · Authors · 2023-08-28
>
> **Note of Thanks**
>  We appreciate your insightful review. We are glad you found the research question we address interesting, and the investigation comprehensive.
>
> **RC: Reviewer's comment**
> **AR: Author's response**
>
>
> _**Reasons to Reject**_
>
> **RC** Evaluation is not rigorous: When using CLIP score to identify if the text correctly describes the object in the image, it is very important to NORMALIZE the score. For example, we first need to calculate the score between the image and all the object categories. And then we normalize all the scores and pick up the maximal one. We can’t directly calculate the score, because even if “toilet paper” has high matching score with the image, there might be another object name that has higher score. The evaluation may hurt the soundness of the method a little bit.
>
> **AR**  _While this is an interesting point, note that in our paper, our emphasis is on evaluating CLIP for its foundational multimodal model's ability to recognize diversity in text-image associations.  As we mentioned in the paper, one topic/label can be represented by a variety of objects. Similarly, one object can be found in multiple topics.  If we were to normalize the scores, it means we would set our experiments to measure precision for object detection or classification, which is different from our objective. Instead, we chose to use the CLIP scores "as is" because we wanted to measure if the objects in the images can be identified/associated with the textual topics they are labeled as_.
>
>
> **RC** May lack more interesting findings: Although the authors provide very comprehensive analysis and in-depth insights, most of them are very similar to what previous works did. For example, in Liu et al., 2021 and Yin et al., 2021, the authors already find that there’s a large performance disparity across Western and non-Western countries. Yin et al., 2023 also quantify the performance gap on Dollar Street. Though it is not going down to the salary level, the basic observation is similar. What I expect is that, besides claiming that the models are biased, please also try to dig out the reason and the ways we can improve. That can make the paper more convincing and attract more people to follow and devote to make the models better. Visually Grounded Reasoning across Languages and Cultures. Liu et al., 2021 Broaden the Vision: Geo-Diverse Visual Commonsense Reasoning. Yin et al., 2021 GIVL: Improving Geographical Inclusivity of Vision-Language Models with Pre-Training Methods. Yin et al., 2023
>
> **AR**  _Thank you for these pointers. While these papers investigate performance disparity along the cultural axis, our work provides insights into differences that exist across income levels irrespective of the country of origin.
> Despite having similar motivations, contributions from Liu et al., 2021, Yin et al., 2021, Yin et al., 2023  focus on cultural concept diversity, where language is a factor that can facilitate an understanding of the disparities. However, as we found in our work, there are also major performance disparities that exist within one culture along the income axis, which has not been accounted for in previous work._
>
> _**Questions for Authors**_
>
> **RC** Why not just use object classification accuracy but use the simple CLIP scores?
>
> **AR** _Object recognition and its downstream applications are indeed apparent areas where equal representation is important. However, there is a danger for multimodal applications such as image captioning, video frame processing, and action recognition to suffer from a limited view of what textual concepts/topics can be associated with visual data.   Therefore, we chose to stick with CLIP similarity scores that measure basic text-image association (rather than focusing on the task of object recognition), to capture the disparities that exist at this fundamental level of associations in the multimodal space. Note also that the performance disparities specific to object recognition have been previously explored by De Vries et al. (2019)._
>
>
> **RC** For visual similarity section, is there any way to extend the analysis to larger-scale?
>
> **AR**  _Yes, following your suggestion we performed an analysis of the visual similarity scores for the 64 (country, income) pairs (where income belongs to the poor or rich quartile). We found that across all topics, the images from poor households are similar to each other, and those from rich households are similar to each other, regardless of the countries these images come from. This is consistent with our findings in RQ3. We plot these findings on a heatmap that you can access here [heatmap](https://anonymous.4open.science/r/Bridging_Digital_Divide-E41C/heatmap.pdf), which shows the visual similarity scores between images from different income levels (poor \$,_ _rich \$\$\$) across all topics, from 64 countries on different continents._
>
>
> 1. De Vries, Terrance et al. (2019). “Does object recognition work for everyone?” In: Proceedings of the
> IEEE/CVF conference on computer vision and pattern recognition workshops, pp. 52–59.
>
> 2. Liu, Fangyu et al. (Nov. 2021). “Visually Grounded Reasoning across Languages and Cultures”. In: Pro-
> ceedings of the 2021 Conference on Empirical Methods in Natural Language Processing. Online and
> Punta Cana, Dominican Republic: Association for Computational Linguistics, pp. 10467–10485. DOI:
> 10 . 18653 / v1 / 2021 . emnlp - main . 818. URL: https : / / aclanthology . org / 2021 .
> emnlp-main.818.
>
> 3. Yin, Da, Liunian Harold Li, et al. (Nov. 2021). “Broaden the Vision: Geo-Diverse Visual Commonsense
> Reasoning”. In: Proceedings of the 2021 Conference on Empirical Methods in Natural Language Processing.
> Online and Punta Cana, Dominican Republic: Association for Computational Linguistics, pp. 2115–2129.
> DOI: 10.18653/v1/2021.emnlp-main.162. URL: https://aclanthology.org/2021.
> emnlp-main.162.
> 4. Yin, Da, Feng Gao, et al. (2023). “GIVL: Improving Geographical Inclusivity of Vision-Language Models
> with Pre-Training Methods”. In: Proceedings of the IEEE/CVF Conference on Computer Vision and Pattern
> Recognition, pp. 10951–10961.

---

### Meta-Review · Area_Chair_PRch · 2023-09-18

**Recommendation:** 5

**Metareview:**

All reviewers agreed that the paper investigates a well-motivated and interesting research question, using comprehensive evaluations. Reviewer Vvi2 raised a soundness concern related to unnormalized use of the CLIP score. However, I agree with the author response that the unnormalized score provides a measure of label-image relevance which is independent of the contrasting images/text chosen, as also used by previous work. I also agree that the paper does make a novel contribution, as previous work on this Dollar Street benchmark does not look at disparities related to income values, and while I agree with reviewer Vvi2 that this result might not be too surprising given prior work, this is still valuable to confirm.

---

### Decision · Program_Chairs · 2023-10-07

**Decision:**

Accept-Main

**Comment:**

All reviewers agreed that the paper investigates a well-motivated and interesting research question, using comprehensive evaluations. Reviewer Vvi2 raised a soundness concern related to unnormalized use of the CLIP score. However, I agree with the author response that the unnormalized score provides a measure of label-image relevance which is independent of the contrasting images/text chosen, as also used by previous work. I also agree that the paper does make a novel contribution, as previous work on this Dollar Street benchmark does not look at disparities related to income values, and while I agree with reviewer Vvi2 that this result might not be too surprising given prior work, this is still valuable to confirm.